# Physiological Changes Across a Sport Season in a Nine-Time World-Champion Triathlete: A Case Report

**DOI:** 10.3390/sports13050140

**Published:** 2025-04-30

**Authors:** Adrian Gonzalez-Custodio, Carmen Crespo, Rafael Timon, Guillermo Olcina

**Affiliations:** Faculty of Sport Science, Universidad de Extremadura, Av. Universidad, s/n, 10003 Cáceres, Spain; adriangc@unex.es (A.G.-C.); ccrespoc@unex.es (C.C.); rtimon@unex.es (R.T.)

**Keywords:** triathlon, elite, physiology, case report

## Abstract

This case report analyses the physiological changes of a nine-time world champion triathlete over a competitive season. The triathlete, aged 34, resumed training after a 3-month injury-related break. The study monitored key physiological variables at three points: pre-season (PRE), base period (BASE), and peak performance (PEAK). The athlete trained an average of 25,000 m swimming, 400 km cycling, and 90 km running weekly. Incremental cycling tests were performed at each stage, measuring power output, oxygen uptake (VO_2_), ventilatory thresholds (VT1, VT2), muscle oxygen saturation (SmO_2_), heart rate, and lactate levels. Results showed significant improvements in relative power output (+37.2% at VT1), VO_2_ max (+12.6%), and body composition (body fat reduced from 10.43% to 7.33%). Heart rate and lactate concentration remained stable, while SmO_2_ showed a greater difference between VT2 and peak performance. The triathlete achieved top-10 finishes in all key events, including a win at the Ironman 70.3 World Championship. The findings suggest that elite triathletes can regain peak performance after injury through structured training, with improvements in ventilatory efficiency and body composition contributing to better competition results. This study provides valuable insights for coaches on the recovery and performance optimization of elite triathletes.

## 1. Introduction

Triathlon is a multidisciplinary Olympic sport comprising swimming, cycling, and running, without breaks during the competition. Currently, there are different competitive distances in triathlon, differentiated into two large groups—competitions with drafting and competitions without drafting. The fundamental difference is that in the cycling segment, riding in a group is allowed. In the case of races with drafting, we will have two main distances, sprint distance (750 m swimming, 20 km cycling, and 5 km running) and the standard or Olympic distance (1500 m swimming, 40 km cycling, and 10 km running). There is scarce scientific literature that analyses elite triathlon, especially international elite triathletes. The scientific literature considers the international elite to be the first 125 athletes in the international ranking of the International Triathlon Union (ITU) [1].

The significance of triathlons is increasing. Every year, the number of international competitions continues to grow, and of course, the specificity of training is more accurate. The physiological requirements of the elite international triathlon have been analysed in some scientific literature [1,2]. While the number of studies in the scientific literature analysing top-class elite athletes is growing [3,4,5] there are more articles that analyse training loads and different training plans. There are some articles that analyse training loads in triathlon, specifically in elite triathletes with an average of 14.7 ± 3.0 h of training [6], or training loads of junior-to elite-class athletes, expressed in different units [7]. None of these articles express the complete physiological profile of an athlete with the characteristics of the subject of this work. There are not enough works that analyse the physiological profiles of elite triathletes. It is important to have information on elite triathletes’ physiological profiles, especially those of the world-class athletes, to provide the information for coaches and triathletes at the elite level to close the gap.

Contemporary triathlons have changed greatly. Events that allow drafting are the most important events in the international calendar. The world triathlon (ITU) calendar is very long and involves many competitions and much travel. It is a challenge for the coaches to organise the training load and peaking. There are some articles which confirm that the cycling segment is one of the most important parts of the triathlon and one of the fundamental aspects in the final result of the competition [8]. Ventilatory threshold 2 or anaerobic threshold (VT_2_) and maximal oxygen uptake (VO_2_ Max) are two of the variables most analysed to predict performance in the elite triathlon [9]. There are some data which analysed these variables in elite triathlon [5,6,10]; however, some of them do not refer to the world-class level, and some of them do not analyse the same characteristics and competitions (paratriathlon). There are studies that have analysed this type of athlete but not in triathlon [11,12]. The aim of this work was to analyse the physiological profile of a nine-time world championship triathlete during a season.

## 2. Materials and Methods

### 2.1. Participant

This study is a case report following the Case Report (CARE) guidelines [13] based on a world-class triathlete [14]. The characteristics of the triathlete are consistent with the scientific literature, which supports their suitability for conducting case studies in athletes [15]. The triathlete had won a total of nine World Championships (2xIronman 70.3, 5xITU Distance, 1xLong Distance Triathlon, and 1xXterra Triathlon) and an Olympic silver medal. The triathlete injured his arm in a bike crash and was unable to participate in the Rio de Janeiro Olympic Games. After 3 months of inactivity due to the injury, he returned to training. The triathlete calendar was focused on the World Triathlon Championship Series and some Ironman 70.3 triathlons. The triathlete was 34 years old. He was 178 cm tall and weighed 73.3 kg. During the season, the triathlete developed an average training regime of 25,000 m of swimming, 400 km of cycling, and 90 km of running per week. The study was approved by the Ethics Committee of the University of Extremadura (ref. 03/2021), and the triathlete signed a written informed consent form before the protocol began.

### 2.2. Study Design

The study aimed to analyse all the seasons of the triathlete, including the preseason. The triathlete’s main objectives were the Ironman 70.3 World Championship and ITU World Championship. The study’s aim was to analyse how physiology variables changed across the season of a top-class triathlete. During the season, the triathlete was measured at three different instances. The first occasion was at the beginning of the season, just after the period of inactivity and before the preseason had started (PRE). The season and first macrocycle of the athlete started on the next day of the first test protocol (PRE). After the first mesocycle of base work, which lasted for 4 weeks, the test protocol was performed again (BASE), and, where the peak performance based on his plan were established, the test protocol was repeated (PEAK). The protocol and the temporary distribution of the objectives and measurements can be seen in Table 1.

### 2.3. Assessment

The measurements were formed by two different protocols: the first was developed using the triathlete’s bicycle in the same laboratory conditions (ambient temperature 22.9 °C ± 1.6 °C, relative humidity 41.5% ± 8.7%), and the second was to analyse his body composition.

The protocol for measuring changes in the physiological variables of the triathlete involved an incremental test conducted until the triathlete reached voluntary exhaustion. The test was performed on the athlete’s bicycle mounted on a roller (Cycleops^®^ The Hammer, Fitchburg, WI, USA) to prevent possible biomechanic interferences. The test started at 100 W and increased 30 W every 3 min until voluntary exhaustion, or the athlete was not able to maintain the watt objective. All the tests performed in the three different instances had the same protocol. The power and cadence were measured with a Quarq D-Zero (Sram, Chicago, IL, USA) power meter installed on the subject’s own bicycle. Before the protocol, the power meter was calibrated following the manufacturer’s instructions. During the protocol, muscular oxygen saturation was measured by near-infrared spectroscopy (NIRS) in the right quadriceps of the triathlete. Lactate was collected 3 min after the triathlete ended the test.

The body composition test was measured by an ISAK-certified investigator, following the ISAK procedure [16].

No problems during the test forced any change to the protocol.

### 2.4. Variables

A total of six skinfolds were measured—abdomen, triceps, subscapular, supra-iliac, quadriceps, and medial calf—by an ISAK-certified investigator following the ISAK procedure. The skinfolds were measured using a picometre in millimetres (Holtain, Crymych, UK), the body mass was measured on a digital scale in kilograms with a precision of 0.1 kg (SECA 769, GmbH & Co., KG, Hamburg, Germany), and height was measured in centimetres using a stadiometer (SECA 769, GmbH & Co., KG, Hamburg, Germany). The body fat percentage (%), fat mass (kg), and fat free mass (kg) were calculated using the Yuhasz equation [17].

The variables measured during the incremental test in the laboratory were ventilatory variables, power output, heart rate, muscular oxygen saturation, and lactate. Ventilatory variables were measured with a Metalyzer 3B+ (Metalyzer 3B, Cortex, Leipzig, Germany) following the manufacturer’s instructions for calibration, and the data were analysed using MetaSoft Studio v.5.16.0. (MetaSoft Studio, Cortex, Leipzig, Germany). The ventilatory variables measured during the test were absolute oxygen uptake (VO_2_), shown as L/min; relative oxygen uptake (rVO_2_), shown as mL/kg/min; pulmonary ventilation (VE), shown as L/min; ventilatory equivalent for oxygen (VE/VO_2_), shown as L/min; and end-tidal partial pressure of oxygen (P_ET_O_2_) and carbon dioxide (P_ET_CO_2_), measured in millimetres of mercury (mmHg). Power output was measured with the power meter, Quarq D-Zero (Sram, Chicago, IL, USA) installed on the bike of the triathlete and connected via Bluetooth to MetaSoft Studio. The power meter was calibrated following the manufacturer’s instructions. We used absolute power output in watts (W), and relative power (W_r_) output was calculated with the weight of the triathlete (W/kg). Power maximum percentage (% Power) was analysed based on the power of the last step of the protocol (%). The heart rate was measured by a Polar H7 Belt (Polar, Oulu, Finland) connected by Bluetooth to MetaSoft Studio. Heart rate reserve based on Karvonen’s formula was calculated and shown in the results [18]. The muscular oxygen saturation (SmO_2_) and total haemoglobin (THb) were measured by MOXY Monitor (MOXY, Hutchinson, MN, USA) in the right vastus lateralis of the quadriceps [19]. SmO_2_ was shown as percentage (%), and total haemoglobin (THb) was shown as (g/dL). Lactate concentration was measured in mmol/L using the Lactate Pro 2 portable blood lactate meter (Arkray, Kyoto, Japan) on micro blood samples drawn from the tip of the index finger, according to the manufacturer’s instructions. The aerobic threshold (VT1) was determined when VE/VO_2_ and P_ET_O_2_ increased with no increase in VE/VCO_2_. The anaerobic threshold (VT2) was determined using the criteria of an increase in both VE/VO_2_ and VE/VCO_2_ and a decrease in P_ET_CO_2_. Two independent observers identified VT1 and VT2 [20]. Oxygen uptake (VO_2_) was considered as maximal (VO_2_ Max) when at least three of the following four criteria were met: (1) a plateauing of VO_2_ (defined as an increase of no more than 2 mL·kg^−1^·min^−1^ with an increase in workload) during the later stages of the exercise test; (2) a HR > 90% of the predicted maximum for their age (220—age); (3) a respiratory exchange ratio (RER) > 1.1; and (4) an inability to maintain the minimal required pedalling frequency (i.e., 60 rpm) despite maximum effort and verbal encouragement. The percentage of maximum oxygen uptake was shown as a part of the results (%VO_2_ Max).

### 2.5. Statistical Analysis

This study is a case study with purely exploratory objectives and does not involve the manipulation and/or randomization of variables, using means as a reference for the data. Additionally, the percentages of change have been calculated based on the different temporal points.

## 3. Results

The results of the different competitions in which the triathlete competed are shown in Table 2. The main objectives of the triathlete were the world championship Ironman 70.3 in Chattanooga, which he could win, and the World Triathlon Championship Series and World Championship of Olympic distance in Rotterdam, in which he could achieve fourth place. In all the other results, the triathlete could come in the top 10; with the consistency of the results obtained throughout the season, they believed that he would be capable of winning the overall classification of the World Triathlon Championship Series.

The physiological variables of the incremental cycling test are shown in Table 3. The results are represented at three different times of the season (PRE, BASE, PEAK).

The percentage of change in power output is expressed in Figure 1, showing relative power and absolute power through the season’s time points. Power output changes are more visible in relative power output (W_r_) based on the changes in body composition. Power output in ventilatory threshold 1 increases much more between pre and base than between base and peak. In ventilatory threshold 2 and maximum oxygen uptake, changes remain constant throughout the measures.

Hear rate and SmO_2_ %changes are shown in Figure 2. Heart rate does not exhibit significant changes over the different time points and remains relatively stable. Except for ventilatory threshold 1, ventilatory threshold 2 and maximum oxygen uptake generally show a downward trend, as does resting heart rate, which also decreased over time.

The observed changes in oxygen consumption show a trend similar to that of power output (Figure 3). The changes are more noticeable in relative values than in absolute values, due to the evolution of body composition. The most notable change is the increase in ventilatory threshold 1 between PRE and BASE. Additionally, a slight improvement in VT2 and maximum oxygen consumption can also be observed between PRE and BASE, which later becomes much more diminished between BASE and PEAK, with only the maximum oxygen consumption value increasing.

Lactate concentrations remain stable between PRE and BASE and decrease very slightly between BASE and PEAK, %Change PRE vs. BASE (4.90%), BASE vs. PEAK (−12.15%), and PRE vs. PEAK (−7.84%)

The last results that this study analysed were the body composition results (Table 4). All the variables were collected at the same time point as the other variables. Weight decreased significantly during all the season (PRE 76 kg vs. BASE 74 kg vs. PEAK 70.5 kg). The skinfold measurements present pronounced differences between the temporary moments (PRE 70 mm vs. BASE 52 mm vs. PEAK 38 mm) with a decrease in the percentage of fat (%) and fat mass (kg).

## 4. Discussion

This study has shown two principal results: the first are the data on the physiological variables of a world-class triathlete as a reference, and the second shows how a world-class triathlete develops the physiological changes necessary to produce peak performance after some time without training due to an injury. The most relevant conclusion we could derive from this work is that an athlete with these characteristics can improve their performance in the first month and then maintain this improvement all season.

We have incorporated a section into the article grounded in recent studies that offer a set of recommendations for conducting case studies involving elite athletes [15]. The characteristics of both the manuscript and the athlete are consistent with the criteria outlined in those studies. Specifically, the athlete falls within Tier 4 and Tier 5 [14], demonstrating a high level of performance, which further supports the appropriateness and relevance of including this case study within the context of elite sports research. Moreover, this work establishes a clear and meaningful connection between the practical and scientific domains, helping to bridge the common gap between these two fields. It aims to promote the practical applicability of scientific knowledge within the context of high-performance sport.

The first finding of this work concerns the physiological variables that were collected during the different tests. There are some works that analyse these data but in different sports, such as cycling, cross-country skiing, athletics, etc. [11,21]. There are some references made to triathlon, but the athlete characteristics are quite different [6]. The VO_2_ Max recorded in the peak performance of this triathlete is in agreement with other investigations that analysed other sports, including cycling [11], marathon running [3], or cross-country skiing [21]. Greater oxygen uptake is related to success in triathlons, especially if the triathlete can maintain higher values of oxygen uptake during long time periods [22]. The results that this study shows in terms of physiological variables are particulary similar in ventilatory parameters. However, differences are observed in values related to power output. Nevertheless, it is important to consider that the material, the methodology used, and the protocols implemented vary among studies.

The scientific literature has analysed how an elite cyclist modifies his or her performance profile after retirement [12]. The results of the study drew similar conclusions to this present work, namely that one of the main changes in an athlete in this situation is in body composition, whereas aerobic performance does not change as much. This conclusion implies that it is essential to consider the importance of monitoring this aspect of the athlete when they have to stop training due to illness or injury.

The scientific literature does not often analyse the physiological changes of an athlete with this characteristic. There are some works which have analysed the training load of a top-class triathlete and the changes that occur in the physiological parameters of this training programme. Some studies confirm that this type of athlete can get into really good shape in a short period of training [6]. The physiological profile of both triathletes analysed in their case study are similar, with similar parameters.

The increase in the VO_2_ Max between off-season and peak performance is one of the main topics of the scientific literature. Specifically, in the triathlon of a top-class athlete, there are some results that confirm an improvement of 7.8% in VT1, 17.8% in VT2, and 19.2% in maximum [6]. It is important to note that the triathlete who has been analysed in this study comes from a period of total inactivity, due to an injury, so some of the references that other articles give could contradict the results of this study. Some studies results present a substantial shift in the VO_2_ [6], but other works do not present changes as marked as this, while some others confirm a 4.5% change in VO_2_ Max between off-season and peak performance in elite runners [23].

Body composition is one of the variables that have been analysed more in terms of sport performance. Sometimes body composition is used as a predictor of success in competition. Studies consistently demonstrate that triathletes with optimal body composition tend to achieve superior competitive results. Athletes with lower body fat percentages and higher lean muscle mass improved their competition results [24]. These studies analysed amateur triathletes. This present study confirms that better body composition is related to better competition results in a world-class elite triathlete.

One of the main characteristics of the contemporary triathlon is the increase in technical courses on the bike section. In the cycling segment of the contemporary triathlon, the many demanding high-power peaks and tactical movements associated with this technical section led to the drafting-legal competition [9]. The difference in SmO_2_ between VT2 and Max is greater in peak than base and pre, so this gives the triathlete more ability to perform more high-intensity intervals, confirmed by other studies [25].

The lactate concentrations in top-class cyclists [11] and swimmers [26] have been studied; in another study, this variable was measured at the same intensity, but in swimming, and the results are similar [6]. To the best of our knowledge, there are no studies that have analysed the lactate concentration in world-class triathletes in maximal aerobic effort in cycling segments. However, lactate concentration is quite low between pre, base and peak, with an improvement in power output, so the performance of the triathlete was developed.

## 5. Conclusions

The results of this study could conclude that a top-class triathlete improves physiological parameters related to VT1 and VT2 after one month of training, but more specific training at high intensity is needed to improve maximal aerobic efforts. It is important to explain that one of the main performance factors influencing competition success are found in the maximal aerobic efforts, so, finally, the triathlete needs to improve in this metabolic zone to achieve the objective.

One of the main limitations of this study is that training load and intensity were not analysed daily, so future studies should analyse this variable to obtain more accurate information that can be applied to methods used to achieve improvement. Another limitation is that the triathlete in this study was measured in the cycling segment—future studies could measure the swimming and running parameters.

This work is a useful tool for elite coaches who have an injured athlete who need to be prepared for an objective. Another important conclusion this study confirms is that an athlete with this characteristic can get back the performance level that has before the injure. This study could be useful to get a physiological profile of a triathlete with this characteristic.

## Figures and Tables

**Figure 1 sports-13-00140-f001:**
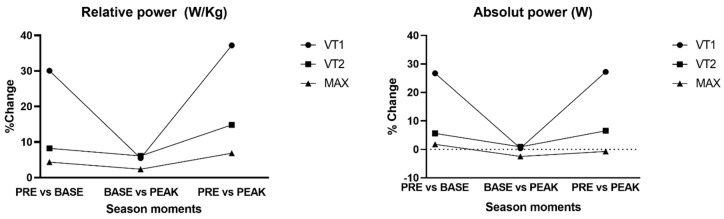
Power output % change between season time points.

**Figure 2 sports-13-00140-f002:**
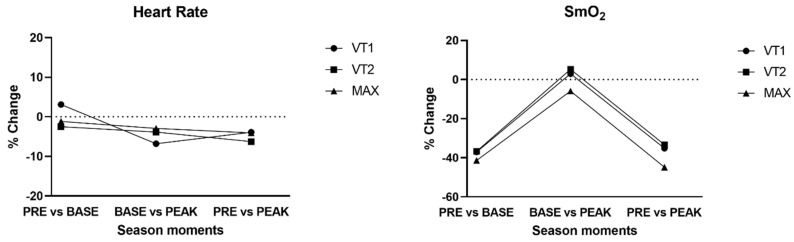
Heart rate and muscle oxygen saturation (SmO_2_) % Change between season time points.

**Figure 3 sports-13-00140-f003:**
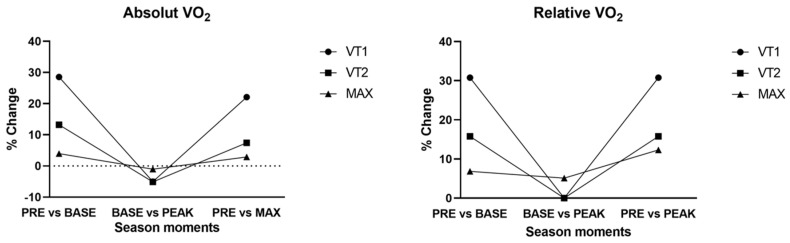
Absolute and relative oxygen consumption and % change between season time points.

**Table 1 sports-13-00140-t001:** Triathlete tests and races.

Place	Event	Date
	Test PRE	1 October 2016
	BW Mesocycle	October–November 2016
	Test BASE	November 2016
Dubai	IM 70.3	27 January 2017
Abu Dhabi	WTS	4 March 2017
Gold Coast	WTS	8 April 2017
Yokohama	WTS	13 May 2017
Hamburg	WTS	15 July 2017
Edmonton	WTS	29 July 2017
Montreal	WTS	6 August 2017
	Test PEAK	16 August 2017
Des Moines	Other	3 September 2017
Chattanooga	IM 70.3 WC	10 September 2017
Rotterdam	WTC-GF	16 September 2017
Bahrain	IM 70.3	25 November 2017

Notes: BW Mesocyle: base work mesocycle IM 70.3, half-distance Ironman; WTS, world triathlon series; WC, world championship; GF, great final.

**Table 2 sports-13-00140-t002:** Triathlete results in international events.

Place	Kind	Date	Position
Dubai	IM 70.3	27 January 2017	1
Abu Dhabi	WTS	4 March 2017	1
Gold Coast	WTS	8 April 2017	4
Yokohama	WTS	13 May 2017	9
Hamburg	WTS	15 July 2017	5
Edmonton	WTS	29 July 2017	6
Montreal	WTS	6 August 2017	1
Des Moines	Other	3 September 2017	1
Chattanooga	IM 70.3 WC	10 September 2017	1
Rotterdam	WTC-GF-WC	16 September 2017	4
Bahrain	IM 70.3	25 November 2017	4

Notes: IM 70.3, half-distance Ironman; WTS, world triathlon series; WC, world championship; GF, great final.

**Table 3 sports-13-00140-t003:** Triathlete’s physiological outcomes in exercise incremental cycling tests across sport season.

	VT1			VT2			MAX		
	PRE	BASE	PEAK	PRE	BASE	PEAK	PRE	BASE	PEAK
Power (W)	202	256	257	322	340	343	398	405	395
Relative power (W/kg)	2.66	3.46	3.65	4.24	4.59	4.87	5.24	5.47	5.60
% Power	51	63	65	81	84	87	100	100	100
Heart rate (bpm)	129	133	124	160	156	150	173	171	166
% Heart rate (Karvonen)	64	68	67	88	86	88	100	100	100
Relative VO_2_ (mL/kg/min)	39	51	51	57	66	66	73	78	82
% VO_2_ Max	53	65	63	78	85	81	100	100	100
VO_2_ (L/min)	2.94	3.78	3.59	4.31	4.88	4.63	5.55	5.77	5.71
SmO_2_ (%)	54	34	35	30	19	20	29	17	16
THb (g/dL)	12.2	12.2	12.6	12.2	12.2	12.6	12.2	12.3	12.5
Lactate (mmol/L)							10.2	10.7	9.4
Resting heart rate (bpm)	46	46	38						

Notes: PRE, pre-season; BASE, base period; PEAK, competitive period; MAX, Maximum; SmO_2_, muscle oxygen saturation; THb, total haemoglobin.

**Table 4 sports-13-00140-t004:** Triathlete’s anthropometric characteristics across the sport season.

	PRE	BASE	PEAK
Weight (kg)	76.0	74.0	70.5
Height (cm)	178	178	178
Abdomen skinfold (mm)	16	10	6
Suprailiac skinfold (mm)	7	6	3
Subscapular skinfold (mm)	10	8	7
Triceps skinfold (mm)	10	8	6
Quadriceps skinfold (mm)	20	14	11
Medial calf skinfold (mm)	7	6	5
Skinfolds sum(mm)	70	52	38
Fat mass (kg)	7.9	6.4	5.16
Fat-free mass (kg)	68.1	67.6	65.3
Fat percentage (%)	10.43	8.68	7.33

Notes: PRE, pre-season; BASE, base period; PEAK, competitive period.

## Data Availability

The data presented in this study are available on request from the corresponding author due to privacy reasons.

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
