# Peer review of "Physiological Changes Across a Sport Season in a Nine-Time World-Champion Triathlete: A Case Report"

_sports, 2025, doi:10.3390/sports13050140_

Round 1

Reviewer 1 Report

Comments and Suggestions for Authors

Dear Authors, I have submitted for review the work of a researcher of an important and essential research topic. It should be emphasized that there are indeed few studies on the evaluation of the training process of professional guilds, including professionals. From my own point of view, I can say (I have completed several full Ironman races myself) that the training process should be subject to detailed control, which can contribute to a more effective training process, and consequently to obtaining the desired sports results. I have a few comments and suggestions in relation to individual sections:
Introduction:
It is worth adding information in this section that training and the training loads used may have different intensity and volume depending on the training period. It will look different in the preparatory period, different in the starting period and different in the transitional period. In the discussed period, the time when the athlete had an injury is also key and it could have and probably affected his overall sports level.

Material and methods:
Please add a graph in this section showing the course of the experiment, including all measurements in relation to time and starts in the period under study. This is included in the table of objectives, but it is also worth showing it in the graph. Please also add information about which start was considered the main start in the macrocycle period and what assumptions were made.

Please also write in this section how the specified cadence value of min. 60 rpm was measured and maintained.
- It should also be emphasized that each exercise test and load selection requires calibration, because the body composition changed, which was confirmed by the results, and the load is selected according to body mass.
- Please also describe how the initial load values ​​were measured and determined in the cycling test, and how load progression was maintained and measured
- Was the load selection in vats always adjusted to the subject's body mass (the body composition showed that it was different depending on the time of measurements)
- Please definitely elaborate on the issue of the training loads used, because the work only provides quantitative criteria for the loads used, i.e. the volume for swimming, cycling and running in km/week.Please also specify whether the % water content was tested, because this may also have an impact on the athlete's body weight and ability to perform training loads

Results
in this section it is worth adding a correlation of at least % of body fat content to the measured parameters, which will allow us to assess whether, as the literature shows, its amount can affect the athlete's performance and, consequently, improve sports results
-it should be noted that body mass, body fat mass and % of body fat content were different depending on the training period
- it would be important to show how training loads were shaped during the measurement period and how they could have an impact on the tested parameters

Discussion
Please expand on the fact that different training loads were used depending on the period, because this is key in the athlete's training. After all, the same loads were not used all the time. They were probably different in the initial preparation phase, and different later on. The type of training and training loads used can significantly affect changes in the parameters tested - please supplement this issue with reference to Your research results.

best regards

Author Response

Dear Reviewer,

First, thank you very much for all your feedback and commentaries, with all this help the work will evolve. The quality of a scientific work is based in the cooperation of all the different parts in the process and all this feedback give to the article a key part of the process. We are going to answer the questions in the same way that the reviewer sends them to us:

Introduction:
It is worth adding information in this section that training and the training loads used may have different intensity and volume depending on the training period. It will look different in the preparatory period, different in the starting period and different in the transitional period. In the discussed period, the time when the athlete had an injury is also key and it could have and probably affected his overall sports level.

  • We have included two references of the training characteristic of other works in the scientific literature between L49-52.

Material and methods:
- Please add a graph in this section showing the course of the experiment, including all measurements in relation to time and starts in the period under study. This is included in the table of objectives, but it is also worth showing it in the graph. Please also add information about which start was considered the main start in the macrocycle period and what assumptions were made.

We had included the base work mesocycle in the temporary distribution L96-L97 and the start of the macrocycle were expressed in line 90-91

- Please also write in this section how the specified cadence value of min. 60 rpm - was measured and maintained.

We have included in line 109, the power meter Quarq D-Zero collects data of power and cadence.

- It should also be emphasized that each exercise test and load selection requires calibration, because the body composition changed, which was confirmed by the results, and the load is selected according to body mass. Please also describe how the initial load values ​​were measured and determined in the cycling test, and how load progression was maintained and measured. Was the load selection in vats always adjusted to the subject's body mass (the body composition showed that it was different depending on the time of measurements)

The load is the same in all the different tests starts in 100W and increased 30W every 3 minutes based in scientific literature protocol, but in the results we have included the relative power of different physiological variables to analyse how body composition affects in global performance of the triathlete.

- Please definitely elaborate on the issue of the training loads used, because the work only provides quantitative criteria for the loads used, i.e. the volume for swimming, cycling and running in km/week. Please also specify whether the % water content was tested, because this may also have an impact on the athlete's body weight and ability to perform training loads

This is one of the main study limitations, we have expressed it in the final part of the conclusion (L285-289), futures studies should improve in that way.

Results

in this section it is worth adding a correlation of at least % of body fat content to the measured parameters, which will allow us to assess whether, as the literature shows, its amount can affect the athlete's performance and, consequently, improve sports results. It should be noted that body mass, body fat mass and % of body fat content were different depending on the training period.

Body composition variables were analysed in results section (L210-L216) and table 4. Discussion section explains how changes in body composition improve sport results (L259-265).

- it would be important to show how training loads were shaped during the measurement period and how they could have an impact on the tested parameters.

Thank you for these suggestions, one of the main limitation of this study is that it does not analyse daily training load, for future studies the scientific literature should analyse these variables to get more information in that way.

Discussion
- Please expand on the fact that different training loads were used depending on the period, because this is key in the athlete's training. After all, the same loads were not used all the time. They were probably different in the initial preparation phase, and different later on. The type of training and training loads used can significantly affect changes in the parameters tested - please supplement this issue with reference to Your research results.

Training load and volume distribution were measured by discipline (swim, bike and run), between L279-280 confirms that the study does not analyse training load day by day. This is one of the main limitations of the study. We can not collect the data of intensity because the coach give us general data of the volume but not specifically the intensity.

We have included a section (L227-236) in the article based on recent studies that provide a series of recommendations for conducting case studies with elite athletes. The characteristics of both the article and the athlete align with those outlined in that work.

Mujika, I., Yamashita, D., & Solli, G. S. (2025). Writing High-Quality Case Studies in Sport Science. International Journal of Sports Physiology and Performance (published online ahead of print 2025). Retrieved Apr 12, 2025, from https://doi.org/10.1123/ijspp.2025-0025

These are the answers to the suggestions that you made to our works. Thank you very much for your time and implication to improve the quality of our work. It is important to us to get this feedback and received new ideas to improve the quality of the work.

Thank you very much

Reviewer 2 Report

Comments and Suggestions for Authors

The aim of this case report was to analyze the physiological profile of a 9-times world championship triathlete during a season after injury. Although the athlete is a world champion and the data is extremely valuable, the issue is that the report does not indicate what was done during the recovery process, which is crucial since the recovery from injury is the primary focus. It is unclear what is novel about this study. Therefore, I have decided to reject the manuscript.

  1. Tracking the recovery process from a three-month injury-related break is essential, yet this is not mentioned in the title.
  2. L18 and L75-76. How was the training volume indicated obtained? This description contradicts the statement that “training load and intensity are not analyzed daily” (L279-280). Without showing the training load during the recovery process, it is impossible to discuss the rate of improvement during recovery.
  3. What kind of injury was it, as mentioned? Without this information, the value of the data concerning the recovery process is unclear. At the very least, details on the type of training and the timing of competition return are needed.
  4. It can be inferred from the ethics committee approval number that the study was approved in 2021. However, the timeline does not match the statement that "the triathlete signed a written informed consent form before the protocol began."
  5. Table 1 should include the date of injury, the start date of training, the return date, the dates and results of each competition, and the dates of testing, presented in a way that makes them clear. Typically, it would be best to organize the information weekly.
  6. Table 1. What does "Kind" mean?
  7. It should probably be "Kyoto" instead of "Kioto."
  8. The way different rates of change are presented as line plots in the figures feels inappropriate. This figure could potentially mislead the readers.
  9. Table 3. Is it "relative VO2" or "relative VO2max"?
  10. Table 3. What does "MAX" mean?
  11. Table 3. The resting heart rate should be placed in the "Heart rate" position. Also, it should be "bpm" instead of "bmp."
  12. It is inappropriate to include speculations that are not sufficiently discussed in the conclusion. Information on the intensity level during the athlete's return to competition is essential.

Author Response

Dear Reviewer,

Thank you very much for all your valuable contributions. All the help and time you have devoted to reviewing and suggesting changes to our work significantly contribute to improving its quality. We therefore consider this aspect to be of vital importance in the development of this scientific article.
Below, we address each of the points you raised, with the corresponding changes reflected in the attached document:

  1. Tracking the recovery process from a three-month injury-related break is essential, yet this is not mentioned in the title.

The athlete has injured his arm with a broken bone which converts the habitual rest between season of 1 month in 3 months. When the protocol began the triathlete was completely recovered and the injury did not affect.

  1. L18 and L75-76. How was the training volume indicated obtained? This description contradicts the statement that “training load and intensity are not analyzed daily” (L279-280). Without showing the training load during the recovery process, it is impossible to discuss the rate of improvement during recovery.

Training load and volume distribution were measured by discipline (swim, bike and run), between L279-280 confirms that the study does not analyse training load day by day. This is one of the main limitations of the study. We can not collect the data of intensity because the coach give us general data of the volume but not specifically the intensity.

  1. What kind of injury was it, as mentioned? Without this information, the value of the data concerning the recovery process is unclear. At the very least, details on the type of training and the timing of competition return are needed.

The kind of injury was an arm injury that does not allow the triathlete trains as the participant section described in material and method (L75-77). The recovery is three months of rest.

  1. It can be inferred from the ethics committee approval number that the study was approved in 2021. However, the timeline does not match the statement that "the triathlete signed a written informed consent form before the protocol began."

The research team that designed this study was also working as part of the athlete’s support team during that period. In 2019, the author began his phD studies, and with all the data already collected, we submitted the project to the ethics committee. However, due to bureaucratic delays, its approval was not granted until 2021.

  1. Table 1 should include the date of injury, the start date of training, the return date, the dates and results of each competition, and the dates of testing, presented in a way that makes them clear. Typically, it would be best to organize the information weekly

Th start of the training is after the pre test as it is specify in the study design section L90-91

  1. Table 1. What does "Kind" mean?

Changed it to event to clarify the meaning of the table

  1. It should probably be "Kyoto" instead of "Kioto."

Changed it

  1. The way different rates of change are presented as line plots in the figures feels inappropriate. This figure could potentially mislead the readers.

The results are all presented in the various tables. The main objective of creating the figure with the percentage changes was to provide an alternative perspective that could facilitate the reader’s interpretation of the data. Some reviewers suggested presenting the figure in this manner; therefore, modifying it at this stage may influence their assessment.

  1. Table 3. Is it "relative VO2" or "relative VO2max"?

Relative VO2 in three differente metabolic zone (VT1, VT2 and MAX) as it expressed in the top of the table

  1. Table 3. What does "MAX" mean?

Maximum, we have included in the table footer

  1. Table 3. The resting heart rate should be placed in the "Heart rate" position. Also, it should be "bpm" instead of "bmp."

Resting heart rate is not the heart rate in ventilatory threshold 1, 2 or maximum, so we have created a different variable to help the readers. We have changed bmp to bpm, thank you.

 We have included a section (L227-236) in the article based on recent studies that provide a series of recommendations for conducting case studies with elite athletes. The characteristics of both the article and the athlete align with those outlined in that work.

Mujika, I., Yamashita, D., & Solli, G. S. (2025). Writing High-Quality Case Studies in Sport Science. International Journal of Sports Physiology and Performance (published online ahead of print 2025). Retrieved Apr 12, 2025, from https://doi.org/10.1123/ijspp.2025-0025

We sincerely thank you for all your contributions to our work. This type of feedback significantly enhances the quality of our study. Having input from external reviewers enriches the research and elevates its overall standard.

Round 2

Reviewer 1 Report

Comments and Suggestions for Authors

Dear authors, thank you for sending the corrected version of the work. My comments and suggestions have been added. The work is interesting and deserves publication, which I will recommend to editors,
it is worth undertaking similar research for amateurs and I would be happy to join the team due to the fact that I myself once started 5 times on the full distance and finished Norseman.
with respect

Author Response

Dear Reviewer,

The scientific process is widely regarded as a collaborative endeavor among the various components of the system. The manuscript we have submitted has been significantly enriched thanks to your valuable feedback and insightful contributions. We are sincerely grateful for your assistance—your corrections have greatly enhanced the quality and precision of our work.

We remain at your disposal for any questions, clarifications, or suggestions you may wish to share with us.

Once again, we extend our heartfelt thanks and remain available for any matter related to the manuscript.

Kind regards,

Reviewer 2 Report

Comments and Suggestions for Authors

As I initially pointed out, this study is not merely describing physiological changes across a sport season. Rather, it focuses on changes induced by retraining following a period of detraining caused by injury-related training cessation. Therefore, I recommend revising the title to explicitly reflect the context of injury recovery and retraining. The current title fails to capture the main focus of the study, which may lead to a misinterpretation of its scope and significance.

Moreover, considering that there is no pre-injury baseline data and no description of RTP protocol, the limitations of the study are substantial. Given these concerns, I maintain my recommendation for the same decision.

Author Response

Dear Reviewer,

Thank you very much for your contributions from the outset. The present study does not aim to assess the rehabilitation process of the athlete’s injury, but rather to conduct a descriptive case study to explore the physiological changes in a subject with such unique characteristics (a nine-time world champion). The fact that the athlete had a prior injury is simply a contextual factor that allows us to identify three distinct moments in their fitness status: an initial period of three months of inactivity (highly unusual for an athlete of this level), a second phase of moderate training, and a final peak-performance stage.

Therefore, the purpose of this study is not to present a rehabilitation protocol or to explain how the injury was treated, but to offer a descriptive analysis of how an elite athlete of this caliber physiologically evolves following a period of inactivity.

For this reason, we have chosen to retain the original title of the study, as we believe it accurately reflects the objective: to describe the physiological progression of a nine-time world champion triathlete over the course of a season.

Once again, we sincerely thank you for your valuable input, which has enriched the manuscript and significantly improved its clarity and precision for both readers and the scientific community.

Thank you very much for your collaboration.
